# Perceived Importance of Public Health Risks in Greece: A Nationwide Survey of the Adult Population

**DOI:** 10.3390/ijerph18168256

**Published:** 2021-08-04

**Authors:** Anna Tzortzi, Melpo Kapetanstrataki, Georgios Rachiotis, Vaso Evangelopoulou, Eleni Leventou, Panagiotis Behrakis

**Affiliations:** 1George D. Behrakis Research Lab, Hellenic Cancer Society, 11521 Athens, Greece; atzortzi@researchlab.gr (A.T.); vevan@researchlab.gr (V.E.); pbehrakis@acg.edu (P.B.); 2Institute of Public Health, The American College of Greece, 15342 Athens, Greece; 3Department of Hygiene and Epidemiology, Medical Faculty, University of Thessaly, 38221 Volos, Greece; grachiotis@gmail.com; 4School of Allied and Public Health Professions, Canterbury Christ Church University, Canterbury CT1 1QU, UK; Elelev99@gmail.com; 5Athens Medical Center, 15125 Athens, Greece

**Keywords:** public health, risk perception, health risk, Greek survey, road accidents, cancer, environment, diet, exercise, contraception

## Abstract

The current study is the first to examine the perceptions of the Greek public towards selected health risks and prioritize perceived risk importance and the needs to be addressed for public health promotion. Participants were asked to consider the individual importance of selected risks and the top three most important risks. Data collection took place on February 2020 in a representative sample of the adult Greek population. Differences between groups were assessed with Chi-square tests. Logistic regression models were used to identify perceptions based on participants’ characteristics. Analysis was conducted in Stata 14, and 1976 adults participated in the survey: 48% male and 52% female. Road accidents, cancer and air pollution and environmental protection were considered the top three most important public health risks. Differences were observed between sexes; females were more concerned regarding the importance of the examined public health risks, and between age groups, younger ages considered STDs and contraception more important than older ages. Finally, non-smokers considered exercise and smoking to be more important than smokers. This is the first study to present and grade the public’s perceptions on the importance of public health risks in Greece. Our study’s prioritization of health risks could aid health authorities in improving and promoting the overall public health in Greece.

## 1. Introduction

Public health quality and promotion requires a multidimensional approach, as determinants include several socioeconomic, environmental, geographical, racial and behavioural factors [1].

The World Health Organisation (WHO) announces the top ten global health threats on an annual basis [2]. While several health threats, such as environmental pollution or cancer, represent an almost universal burden, others, such as EBOLA, are only met in certain areas, indicating the need for a locally tailored approach.

Moreover, behaviour depends on the individual’s multifactorial formed subjective perception of health risk and disease, greatly challenging the success of public health policies [1].

Life expectancy in Greece is longer by 6 months compared to the EU average; however, it follows a slower increase [3]. Several public health threats are associated with specific lifestyle choices of the Greek public; as published in the State of Health in the EU-Greece Country health profile 2019, smoking, alcohol consumption and unhealthy diet were associated with 40% of mortality, including an increase in mortality due to lung, pancreatic and colorectal cancer, chronic respiratory diseases and diabetes [3].

A previous study has shown a 44% HPV prevalence among 18–71 year-old gynaecology outpatients [4], while according to a review on HIV epidemiology in Greece, 40% of young adults aged 15–24 years old reported no use of condoms, even though they admitted having more than one sexual partner [5], thus placing sexually transmitted diseases (STDs) and contraception among public health risks in Greece.

Additionally, vaccination coverage differs among age groups, with a satisfactory vaccination coverage [6,7] in children and a low vaccination coverage regarding life-threatening illnesses in seniors [8].

Behavioural risk factors for public health include conduct in and around water, as well as driving and internet browsing practices. In Greece, drowning leads to 400 fatalities annually [9], while road accidents, although showing a reduction in recent years (10,848 in 2017 vs. 23,001 in 2000), still lead to more than 700 fatalities every year [10,11,12].

Regarding internet browsing, in a study conducted in 2018 in 14,000 pupils in Greece, 69% of children admitted to daily internet use and 83% of children to unsupervised internet browsing. Additionally, one out of five children admitted to have been the victim of some kind of online harassment at some point [13].

Socioeconomic factors caused by the economic crisis in Greece over the past 10 years and subsequent unemployment has led to an increase in mental health disorders, including depression [14,15]; the one-month prevalence rate for major depression doubled in 2011 (8.2%) compared to 2008 (3.3%) [16].

Public health quality in Greece is also dependent on environmental factors, including air pollution and natural disasters. The high urban pollution in Greece [17] is associated with adverse health effects, including cardiovascular and respiratory disorders [18]. Natural disasters happen on an annual basis, including wildfires, flooding and earthquakes; due to its geographical location, Greece is susceptible to earthquakes, experiencing 30 daily earthquake tremors [19].

The abovementioned threats are usually addressed by public health policies designed on a multinational, European or global level. However, the burden [20] in terms of morbidity, mortality and economy is still considerable, indicating the urgent need for an alternative approach, both horizontal and population tailored, in designing successful public health policies.

Public perceptions of risk and risk reduction are of importance in public health [21]. In fact, hazards and risks are not necessarily viewed equally by the public.

To the authors’ knowledge, there is a literature gap in studies that have examined the perceptions of the Greek public, as well as the perceptions of the public in European countries, in relation to the most important health risks.

Therefore, the aim of the present study was to define the major health threats for the Greek population, study the perceptions of the public towards those threats, prioritize the needs and indicate the crucial areas for intervention that will ultimately improve the overall quality of public health in Greece.

## 2. Materials and Methods

A cross sectional survey based on a nationally representative sample of the adult Greek population was conducted in 2020 to assess the public’s perceived health risk in relation to selected risks for public health in Greece.

### 2.1. Sample

Adult Greek residents, 18 years old and above at the time of the survey, were enrolled. Data collection was performed during the first week of February 2020.

### 2.2. Methodology and Data Collection

The sample was representative of the adult Greek population according to national census data of 2011, published by the Hellenic Statistical Authority, by sex, age and residence. Data collection was conducted using computer-assisted telephone interviewing (CATI). A 95% sampling error was calculated as ±2%, assuming random sample methodology. Data collection was performed by Kapa Research, an independent organization specializing in population-based surveys, a member of SEDEA (Association of Opinion and Market Research Companies) and of ESOMAR (the World Association of Opinion and Marketing Research Professionals) and complies with their codes of conduct.

Informed consent was obtained individually for each participant verbally from the abovementioned organization prior to the collection of the data, while a fully anonymized dataset with no personal information was later passed on to our team for analysis and results dissemination.

On the onset of the call, participants were informed about the organization conducting the survey, the scope and duration of the survey, and how their data would be handled (fully anonymized and only for statistical purposes) and were asked to give their verbal consent to participate in the survey. Those who opted in continued with the survey, while those who refused to participate were thanked for their time and the call was ended.

### 2.3. Health Risks

Topics were chosen based on the European Observatory on Health Systems and Policies, 2019, regarding the public health status in Greece [3], on Greece: Health System Review regarding the burden of disease [22] and on international surveys on the public’s perception on health risks [23,24].

### 2.4. Questionnaire and Definitions

Participants answered two questions based on the identified public health risks: “How important do you consider the following topics related to public health in Greece?”, giving the option to answer using a scale of “Very important”, “Moderately important”, “Slightly important”, “Not important” and “Don’t know/No answer” and “Which of these topics do you consider the most important?”, giving the option to choose up to three topics. The topics included cancer, road accidents, diet and obesity, vaccination, smoking, STDs, depression, contraception, alcohol abuse, physical exercise, natural disasters (e.g., earthquakes, fires, flooding), air pollution and environmental protection, safety in and around water (e.g., sea/beach/pool/water-sports) and safe internet browsing. The novel coronavirus disease (COVID-19) was not included in the questionnaire as the survey was conducted prior to its emergence in Greece. Additional information collected was participants’ age, sex and level of education.

### 2.5. Statistical Analysis

Analysis was performed for the sample as a total as well as for different subgroups defined by sex and age. Additionally, in cases where smoking status could affect people’s view of certain public health risks, analysis was further performed for smokers and non-smokers. The Chi-square test was used to assess differences between groups. Results are presented as frequencies or percentages.

Multivariate ordered logistic regressions were performed for each public health risk. The proportional odds assumption was tested with the Brant test for all models. For models that the proportionality was not met, generalized ordered logistic regressions were performed instead. Additionally, for the top three most important public health risks identified by the univariate analysis, multivariate logistic models were performed. Explanatory variables were age and sex in all models, irrespective of their statistical significance, because age and sex are known confounders. Furthermore, the interaction of age and sex as well as the variables showing the smoking status and educational level were considered for inclusion in the final models. The final models were concluded with backward elimination procedures. The results of the models are described in the methods section, while the tables are presented on Appendix A and Appendix B.

Statistical significance for all tests was set at *p* < 0.05. All *p*-values presented are two-tailed. Analysis was performed in Stata 14 (StataCorp. 2015. Stata Statistical Software: Release 14. College Station, Texas: StataCorp LP, TX, USA).

## 3. Results

### 3.1. Total Population

A total of 1976 adults, 48% male and 52% female, participated in the survey. Participants represented each age group, as shown on Table 1.

Regarding public health risks, cancer was considered very important by most participants (81%) followed by road accidents (79%), while air pollution and environmental protection were deemed very important by 67% of participants. Vaccination followed, with 64% of participants considering it very important, while natural disasters (such as earthquakes, flooding, etc.) were considered very important by 60% of the respondents. Other public health risks, including STDs, smoking, diet and obesity and depression, were considered very important by 51–59% of participants, while safe internet browsing, alcohol abuse, contraception, exercise and safety in and around water were deemed very important by 40–48% of participants (Figure 1).

When asked which three public health risks they considered most important, road accidents (59%), cancer (56%) and air pollution and environmental protection (32%) were the top three that stood out (Figure 2).

### 3.2. Analysis by Sex

Differences were observed between sexes regarding the importance of public health risks, with females being more concerned about the public health risks examined. Statistically significant differences were observed between sexes for the majority of public health risks, including cancer (84% of females vs. 77% of males, *p* = 0.002), road accidents (82% of females vs. 75% of males, *p* = 0.002), air pollution and environmental protection (73% females vs. 60% males, *p* < 0.001), vaccination (70% females vs. 57% males, *p* < 0.001), natural disasters (67% females vs. 51% males, *p* < 0.001), STDs (67% females vs. 51% males, *p* < 0.001), diet and obesity (56% females vs. 48% males, *p* = 0.001), depression (58% females vs. 44% males, *p* < 0.001), safe internet browsing (56% females vs. 39% males, *p* < 0.001), alcohol (54% females vs. 37% males, *p* < 0.001), contraception (55% females vs. 36% males, *p* < 0.001) and safety in and around water (46% females vs. 33% males, *p* < 0.001) (Figure 3, Table 2).

Regarding which public health risks they considered most important, both sexes agreed that road accidents were the most important (60% of males vs. 58% of females) followed by cancer (55% of males vs. 57% of females), both non-statistically significantly different. Air pollution and environmental protection was ranked 3rd in importance by both sexes but presented a statistically significant difference (*p* = 0.007) between males (29%) and females (34%) (Table 3).

### 3.3. Analysis by Age

Statistically significant differences were observed between individuals of different age groups, as did their opinion on how important several public health risks are. In more detail, road accidents were deemed very important by most people in each age group, ranging from 73% among 55–64 year olds (y.o.) to 83% among 25–34 y.o. and 35–44 y.o., the differences being highly statistically significant (*p* = 0.02). Opinion on the importance of diet and obesity differed among age groups as well, with the 18–24 y.o. (42%) and 75+ y.o. (33%) not considering it as important as individuals in the other age groups (≥50%), which was a highly statistically significant difference (*p* = 0.003). Marginally statistically different (*p* = 0.047) was individual opinion on exercise among different age groups, with fewer individuals 18–24 y.o. considering it very important (34%) compared to the other age groups (40–47%). Opinion on the importance of STDs declined with age, with 73% of 18–24 y.o. considering it very important compared to <70% among participants of the other age groups, and especially lower among those aged 55–64 y.o. (53%), differences that were highly statistically significant (*p* = 0.003). Opinion on smoking differed among age groups (*p* = 0.023), with older individuals considering it more important than the younger ones; specifically, 73% of the 75+ y.o. and 61% of the 65–74 y.o. considered smoking very important; in the rest of the age groups, smoking importance ranged between 52% and 58%, while the lowest, 47%, was reported by the 18–24 y.o. Finally, statistically significant differences (*p* < 0.001) were observed regarding opinion on contraception between different age groups, with importance decreasing with increasing age; 55% of the 18–24 y.o. and 51% of the 25–34 y.o. considered it very important, while in the other age groups <50% considered contraception very important. In particular, 44% of 65–74 y.o. and 31% of the 75+ y.o. considered it very important (Figure 4, Table 4).

The top three reported most important public health risks were the same in all age groups, with no statistically significant differences. Natural disasters were considered one of the most important public health risks by more individuals 18–24 y.o. (32%) and 35–44 y.o. (33%) than in the other age groups, differences that are highly statistically significant (*p* = 0.01). Additionally, vaccination depicted highly statistically significant differences (*p* = 0.009) between age groups regarding most importance, with 26% of the 18–24 y.o. and 75+ y.o. considering it one of the most important public health topics compared to the lower proportion reported by the other age groups. STDs and contraception depicted highly statistically significant differences (*p* < 0.001 and *p* = 0.002, respectively) between age groups, with more younger individuals considering them to be one of the most important public health risks than the older ones. Finally, 38% of 75+ y.o. considered smoking one of the most important public health risks, a much higher percentage compared to the other age groups, where 19–26% considered it one of the most important risks, which is a difference that is highly statistically significant (*p* = 0.012) (Table 5).

### 3.4. Analysis by Smoking Status

Differences were depicted between the importance of several public health risks and smoking status of participants. In more detail, highly statistically significant differences (*p* = 0.002) were observed regarding the importance of diet and obesity, with 47% of smokers vs. 54% of non-smokers and 61% of respondents with an undeclared smoking status considering diet and obesity to be very important. Similarly, highly statistically significant differences (*p* = 0.025) were observed regarding exercise, with 40% of smokers considering exercise very important compared to 47% of non-smokers and 42% of those with undeclared smoking status. Finally, the view on smoking as a health risk was highly statistically significantly different between smoking status (*p* < 0.001), with 34% of smokers considering it very important compared to 65% of non-smokers and 28% of those with undeclared smoking status (Figure 5, Table 6).

Differences were observed on what participants considered as one of the most important public health risks depending on their smoking status. In particular, highly statistically significant differences (*p* = 0.011) were observed regarding diet and obesity, with 20% of non-smokers considering it one of the most important public health risks compared to 14% of smokers and 17% of individuals with undeclared smoking status. Additionally, highly statistically significant differences (*p* = 0.002) were observed regarding depression, with 24% of smokers and 25% of individuals with undeclared smoking status considering depression one of the most important public health risks compared to 18% of non-smokers. Finally, highly statistically significant differences (*p* < 0.001) were observed regarding smoking, as 28% of non-smokers considered it one of the most important public health risks compared to 10% of smokers and 17% of those with undeclared smoking status (Table 7).

Analysis was also performed for education levels and the importance of public health risks, which did not yield statistically significant differences between different education levels. Results are therefore not presented.

### 3.5. Multivariate Analyses

#### 3.5.1. Perceived Importance of Public Health Risks

Modelling of the public risk factors are in line with the results observed by the univariate analysis; women in general are more likely to be concerned for the majority of the public health factors compared to men, adjusted for age. An interaction was observed in the model of air pollution and environmental protection between age and sex, showing that the effect of sex on the concern regarding air pollution and environmental protection differs across the age groups. In the models for STDs and contraception, there were highly statistically significant differences between age groups, adjusted for sex, with younger individuals being more concerned compared to individuals of older age groups. In the model for diet and obesity, differences were highlighted between age groups, with older individuals being more concerned compared to younger individuals. Similarly, individuals of a higher education were more concerned compared to individuals of an education up to secondary, adjusted for age, sex and educational status. Differences between age groups were also observed in the models for alcohol and exercise (Table A1). In the model for cancer, adjusting for sex and age, it was observed that females were more likely to be very concerned than to be moderately or not concerned compared to males. Finally, in the model for smoking, adjusting for age, sex and smoking status, it was observed that non-smokers were more likely to be more concerned for smoking as a public health risk compared to smokers, while older individuals were more likely to be more concerned compared to younger ones (Table A2).

#### 3.5.2. Top Three Most Important Public Health Risks

Multivariate logistic regression models for the top three most important public health risks highlighted differences in prioritization between sexes and age groups. In the model for road accidents, adjusted for sex, individuals aged 45–54 years old were 1.5 times more likely to consider road accidents among the most important public health risks compared to 18–24 y.o. (*p* = 0.04), while individuals aged 55–64 years old and 65–74 years old were 1.7 (*p* = 0.01) and 1.6 (*p* = 0.02) times more likely compared to 18–34 y.o., respectively. In the model for cancer, adjusted for sex, individuals aged 35–44 years old and 55–64 years old were 1.5 times more likely to consider cancer one of the most important public health risks compared to 18–24 y.o. (*p* = 0.048 and *p* = 0.04, respectively). Finally, in the model for air pollution and environmental protection, adjusted for age, females were 1.3 times more likely to consider air pollution and environmental protection among the most important public health risks compared to males (*p* = 0.006) (Table A3).

## 4. Discussion

The present study is the first to examine the public’s opinion on a comprehensive list of lifestyle choices and selected diseases that determine public health quality in Greece. Moreover, it is the first study to rank public health threats by the public’s perceived risk importance and to reveal the varying patterns of public health risk perceptions by sex, age and smoking status. Additionally, reverse reading of the current results, i.e., risks perceived by the public as of lower importance, highlights the existing gaps in public health protection that need to be tackled.

We found that females recorded a higher perception of public health risks in comparison to males. This is in line with accumulated evidence that risk perception is differentially distributed across sexes. In particular, females are more concerned about risks than males [25]. Further, it has been reported that females have also been found to show concern about environmental impacts on their health and implement behaviour changes because of these perceived impacts [26].

Road accidents, cancer and air pollution and environmental protection were the three major concerns, with road accidents considered to be the most important public health risk among respondents. This finding is in line with Greece holding one of the highest places for road accidents in the EU, with 65.2 fatalities/million inhabitants in 2018, well above the estimated EU average of 52.5 fatalities/million inhabitants. Fatalities from passenger car accidents are only slightly elevated compared to the EU average (24.9/million inhabitants in Greece vs. 23.5 in the EU). Motorcyclist fatalities, however, are much higher, because Greece experiences almost double the EU average (17.7 fatalities/million Greek inhabitants vs. 7.9 fatalities/million EU inhabitants) [27]. A likely explanation could be the higher motorcyclist population in Greece (7%) possibly due to the mild climate and increased traffic congestion similarly to Italy (8% motorcyclists) and in contrast to the other EU countries [28]. The fact that 40% of road accidents occur to 25–49-year-old drivers [10], might explain why road accidents were reported first in importance by the 25–44 y.o., and among the most important public health risks by those aged >45 y.o. compared to 18–24 y.o.

The second leading cause of death globally is cancer, with an estimate of more than 9 million deaths in 2018 worldwide [29]. While cancer is not the leading cause of death in Greece, respondents ranked cancer as the second most important public health risk, which is possibly a reflection of the increased mortality due to lung, pancreatic and colorectal cancer observed since 2000 [3]. Smoking, however, a factor causally linked to lung cancer, was not considered among the top three most important public health risks by the Greek public, a finding that could be interpreted considering two different aspects. Firstly, the reduced smoking prevalence observed in Greece during the past decade and the adherence to the smoking ban law in recent years were factors that likely helped shape a “problem solved” public perception regarding smoking [30]. Secondly, it highlights the need for enhanced education, communication and awareness and the crucial role to be undertaken by physicians [31], health care providers and educators [32] for successful smoking prevention and cessation.

Air pollution has been associated with cardiovascular and respiratory disorders, including an increased incidence of lung cancer. Data from 17 European studies including Greece showed that increased concentration of particulate matter with a diameter less than 10 μm (PM10) is associated with increased lung cancer incidence; furthermore, the PM10 concentration measured in Athens, also associated with an increased lung cancer risk [33], may explain why air pollution and environmental protection were third among the three most important public health risks in the current study. Additionally, recent research explores the role of urban air pollution on COVID-19 susceptibility through (a) the exacerbation of chronic respiratory and cardiovascular disorders, already a risk factor for COVID-19 and (b) the possible role of microparticles in serving as a vehicle for the virus, a concept yet to be confirmed [34]. Furthermore, the complex environmental effect of human activities is considered to be a crucial factor in:The transmission and new emergence of infectious diseases [34]; the highly rated air pollution and environmental protection in the present study reveals the increased environmental awareness of Greek citizens, which is also supported by Drimili et al., who showed a high rate of recycling practice by Athens’ residents [35].The increased incidence of natural disaster phenomena that has been associated with climate change and global warming; the frequent earthquake, wildfire and flooding events in Greece possibly explain why one in four respondents considered natural disasters to be a major public health risk. Between 2010 and 2016, 85% of the total burned area in Europe was due to fires in Greece, Spain, Portugal, France and Italy, with over 40,000 fires per year [36], whereas in 2018 a catastrophic fire led to 102 fatalities in Attica, Greece [36]. Despite its mild temperate climate, Greece is not spared from flooding; from 2002 to 2013, 22 flooding events occurred in Greece [37], while during the period 1970–2010, 54 events led to 151 fatalities [38]. In a 2018 survey, the Greek public ranked flood risk third in importance among natural disasters, following earthquakes and wildfires, with females considering flood risk to be more important than males [39], a difference also supported by current study results.

While cancer was reported as one of the most important public health risks, the leading causes of death in Greece are stroke and ischaemic heart disease [3]. Smoking, unhealthy diet, lack of exercise and excessive alcohol consumption are documented risk factors for the development of both of the abovementioned disorders [40,41,42,43], all of which are modifiable life-style choices, yet they were not considered as important by the Greek public in the current study; these findings are in line with Ntaios et al., who found that, despite moderate stroke awareness, perceived stroke risk was low [44]. The importance of diet and obesity, exercise and smoking in the current study was perceived differently according to smoking status, with non-smokers and men considering these factors more important. The current study finding is in line with previous studies showing that non-smokers are more prone to follow a healthier lifestyle [45] and that only 15% of smokers consider themselves as being at high risk of stroke [44], supporting the concept that a perception of low health risk lowers the “protection motivation” and “protection action”, thus preventing the individual from adopting the appropriate lifestyle changes [46].

In Greece, paediatric vaccination coverage is overall high, with lower coverage observed among the children of the minority population [47]. Regarding the adult vaccination coverage for measles, rubella, varicella, hepatitis A and hepatitis B, a study in male Air-Forces personnel in Greece found that only 11% of respondents were fully vaccinated against all five diseases, although vaccination coverage was high for the majority of them individually [48]. As showed by the present study, females perceive vaccinations as being of higher importance than males, a finding in line with a previous study showing increased influenza vaccination coverage in women in Greece [32]. Women, in general, may experience the benefits of vaccination more clearly as they develop a higher response to vaccines than males [49]. In contrast, a survey on the intention of healthcare workers to get vaccinated against the pandemic H1N1 showed a higher intention for males [50], an example of how perception of a high health risk, such as was the case with the 2009 pandemic, increases the “protection motivation” and consequently the likelihood of “protection action” [46].

According to the World Health Organization [51], depression is the leading cause of disabilities worldwide and can lead to suicides. Austerity measures, introduced in Greece in 2011, increased the number of suicides in both males and females [52], showing that mental health is a more important public health risk than is perceived by the public. As mental health disorders are still socially stigmatized, patients are less likely to acknowledge the problem and seek care, leading to diagnostic underestimation [53], which in turn may possibly explain the current findings of a relatively low consideration of it as a major public health threat.

STDs and contraception, although not considered among the most important public health risks, were individually considered very important and, as expected, there were depicted age related differences regarding the perceived importance of these risks, which is most likely explained by the increased sexual activity and child bearing potential that characterize younger ages.

Safe internet browsing was very low ranked as an important public health risk by the participants of the current study. In a previous study including 5590 students, 7.3% declared having experienced cyberbullying as victims, and 6.6% as perpetrators [54]. Internet safety represents a serious public health risk that challenges modern societies; the low digital literacy and media education in Greece [55] only add to this challenge, and this calls for capacity building in the educational, regulatory and protection sectors.

According to the WHO, drowning is the third cause of unintentional death worldwide [56]. Greece, a country literally immersed in water, experiences a significant number of drowning events every year [9] and is placed above the EU average and ranks fourth among the other EU countries [45] for accidental drownings. Yet, only 1% of current study respondents considered safety in and around water one of the top three most important public health risks, highlighting the gap in risk awareness, swimming skill learning and safety skills training.

### Limitations

Current results are subject to some limitations. Our study has a cross sectional design and it is impossible to decipher the causal sequence. Furthermore, our study is questionnaire-based, and information bias may have occurred. Finally, as our study was CATI, individuals without a permanent landline, such as refugees or non-documented migrants, could not be included.

## 5. Conclusions

This is the first study to present the public’s perceptions and grading of the importance of a number of timely public health risks in Greece and the first to identify the top three most important public health concerns, which, in order of importance, are: road accidents, cancer and air pollution and environmental protection. Furthermore, the public health risks that were perceived as less important by the public identified the gaps in awareness and education, highlighting a new field for public health scientists to delve into, for campaigns to focus on and for policy makers to target in the process of designing an updated National Public Health Policy, tailored to the current needs of the Greek Society. Our study’s prioritization of health risks could serve as a tool to aid health authorities in improving and promoting the overall public health in Greece. Therefore, our results can inform an evidence-based policy relating to public health risk management in Greece.

## Figures and Tables

**Figure 1 ijerph-18-08256-f001:**
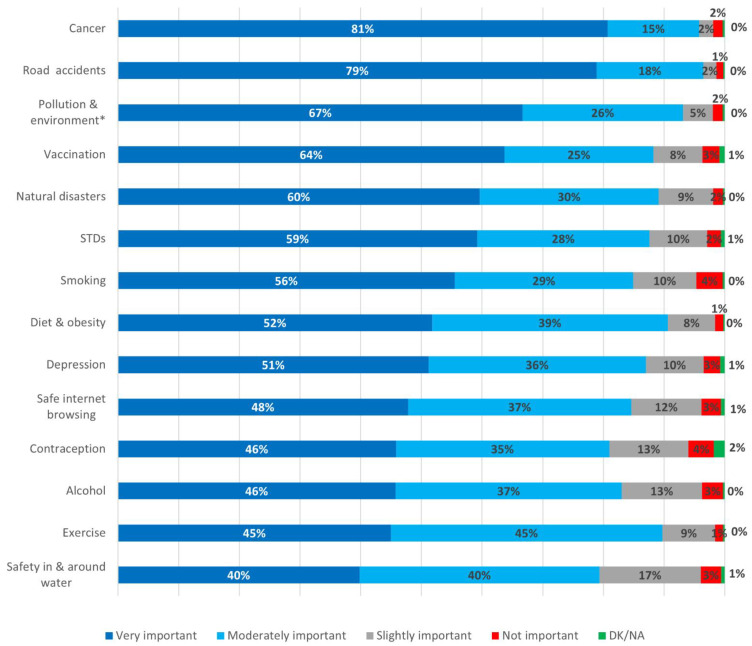
Importance of public health risks; * Pollution and environment stands for air pollution and environmental protection; STDs, Sexually transmitted diseases; DK/NA, Do not know/No answer.

**Figure 2 ijerph-18-08256-f002:**
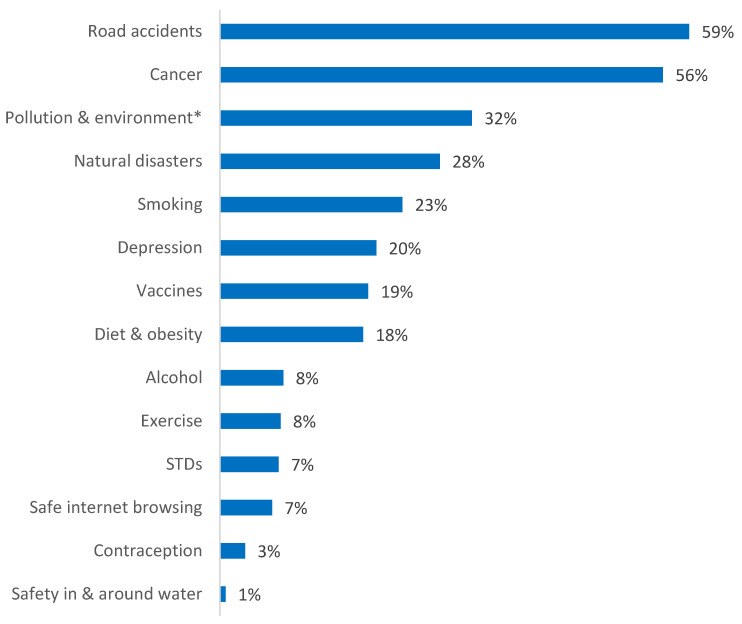
Most important public health risk (up to 3 choices per person); * Pollution and environment stands for air pollution and environmental protection; STDs, Sexually transmitted diseases.

**Figure 3 ijerph-18-08256-f003:**
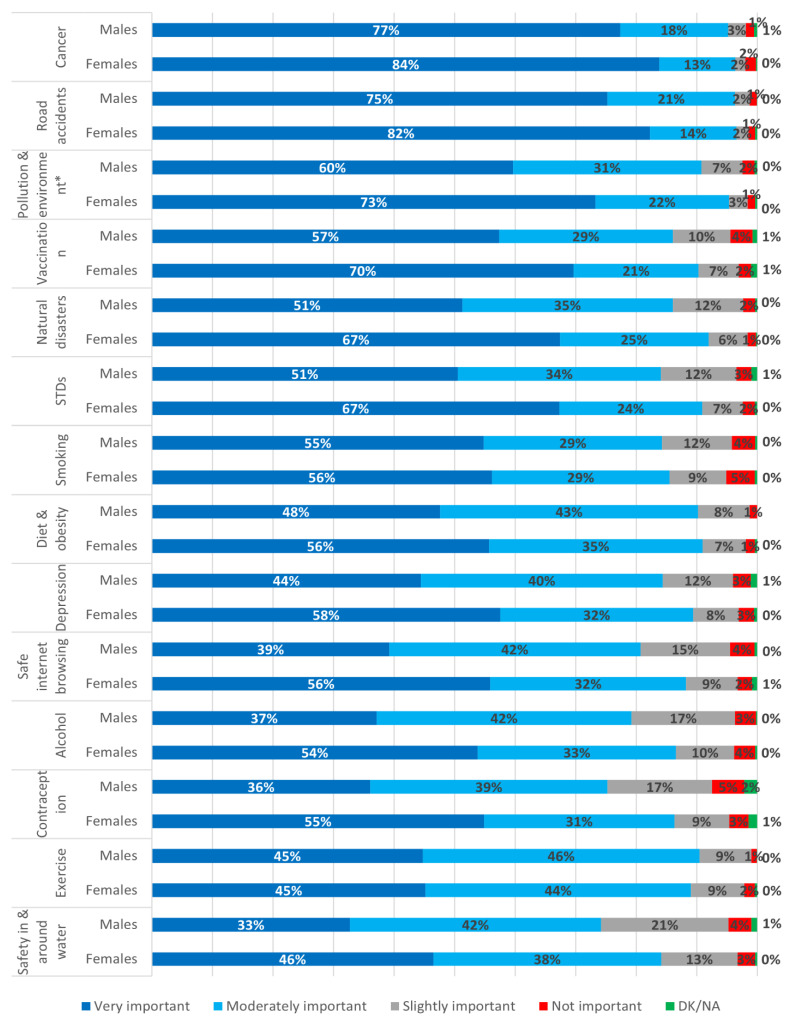
Importance of public health risks by sex; * Pollution and environment stands for air pollution and environmental protection; STDs, Sexually transmitted diseases; DK/NA, Do not know/No answer.

**Figure 4 ijerph-18-08256-f004:**
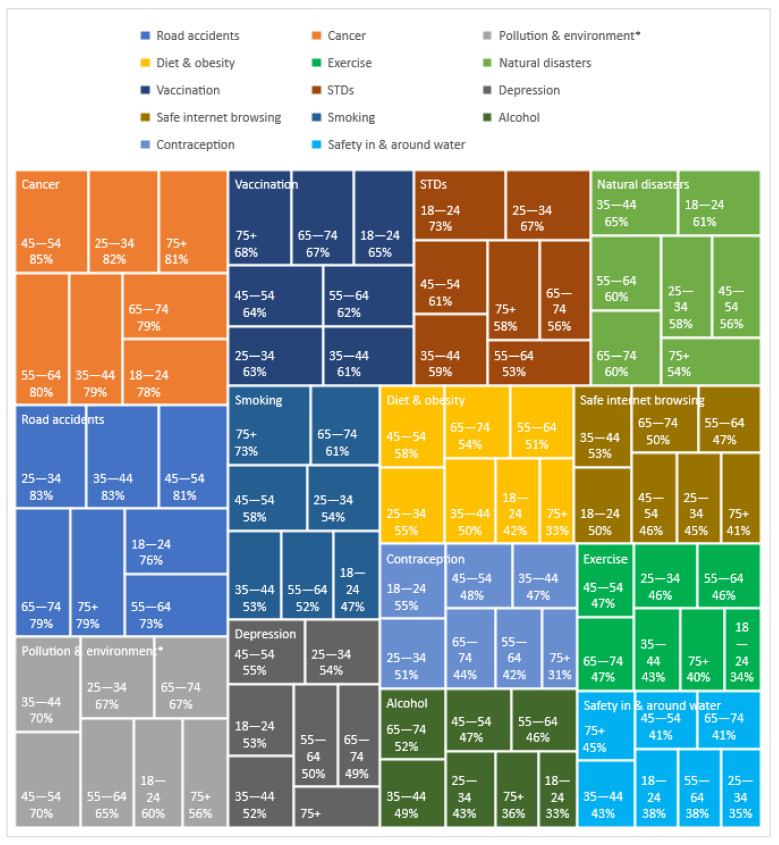
Importance of public health risks by age (showing only “Very important” category); * Pollution and environment stands for air pollution and environmental protection; STDs, Sexually transmitted diseases.

**Figure 5 ijerph-18-08256-f005:**
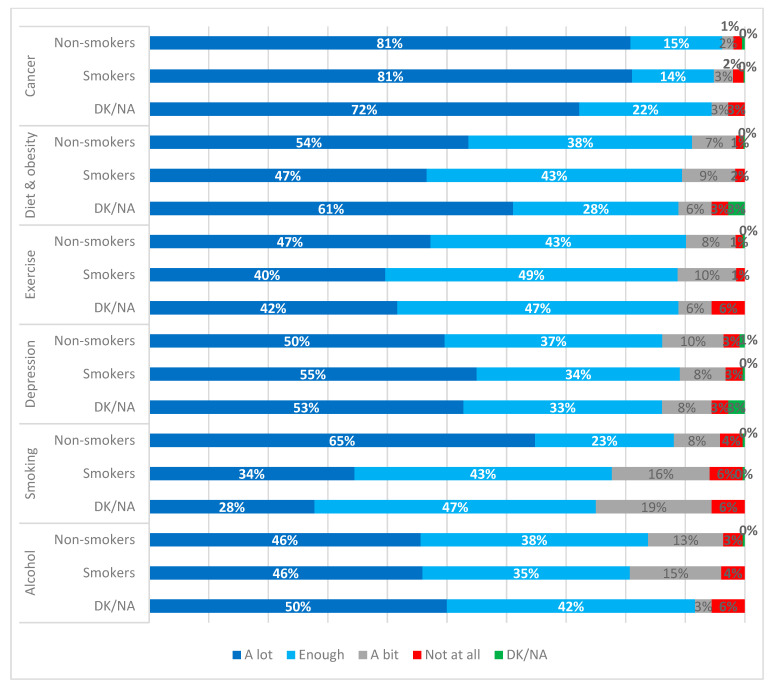
Importance of public health risks by smoking status.

**Table 1 ijerph-18-08256-t001:** Sample characteristics.

Variable	*n* = 1976
Sex	
Male	48%
Female	52%
Age	
18–24	7%
25–34	14%
35–44	16%
45–54	19%
55–64	24%
65–74	17%
75+	4%
**Education**	
Up to secondary education	50%
Higher education	50%

**Table 2 ijerph-18-08256-t002:** Chi-square tests regarding the importance of public health risks by sex (importance defined as “Very important”, “Moderately important”, “Slightly important”, “Not important”, “Do not know/No answer”).

Public Health Risks	*p*-Value
Cancer	0.002
Road accidents	0.002
Air pollution and environmental protection	<0.001
Vaccination	<0.001
Natural disasters	<0.001
STDs	<0.001
Smoking	0.529
Diet and obesity	0.001
Depression	<0.001
Safe internet browsing	<0.001
Alcohol	<0.001
Contraception	<0.001
Exercise	0.265
Safety in and around water	<0.001

STDs, Sexually transmitted diseases. Note: Table 2 presents *p*-values corresponding to Figure 3.

**Table 3 ijerph-18-08256-t003:** Most Important public health risk by sex.

Public Health Risk	Males	Females	*p*-Value
Road accidents	60%	58%	0.488
Cancer	55%	57%	0.466
Air pollution and environmental protection	29%	34%	0.007
Smoking	27%	19%	<0.001
Natural disasters	25%	30%	0.005
Depression	20%	20%	0.893
Diet and obesity	20%	16%	0.045
Vaccination	17%	20%	0.102
Exercise	10%	5%	<0.001
Alcohol	8%	8%	0.963
STDs	6%	8%	0.071
Safe internet browsing	6%	7%	0.295
Contraception	3%	4%	0.165
Safety in and around water	1%	1%	0.345

STDs, Sexually transmitted diseases.

**Table 4 ijerph-18-08256-t004:** Chi-square tests regarding the importance of public health risks by age (importance defined as “Very important”, “Moderately important”, “Slightly important”, “Not important”, “Do not know/No answer”).

Public Health Risks	*p*-Value
Road accidents	0.02
Cancer	0.093
Air pollution and environmental protection	0.176
Diet and obesity	0.003
Exercise	0.047
Natural disasters	0.139
Vaccination	0.221
STDs	0.003
Depression	0.335
Safe internet browsing	0.716
Smoking	0.023
Alcohol	0.056
Contraception	<0.001
Safety in and around water	0.859

STDs, Sexually transmitted diseases; Note: Table 4 presents *p*-values corresponding to Figure 4.

**Table 5 ijerph-18-08256-t005:** Most important public health risk by age (up to 3 answers per participant).

Public Health Risks	18–24	25–34	35–44	45–54	55–64	65–74	75+	*p*-Value
Road accidents	50%	56%	56%	61%	63%	62%	58%	0.072
Cancer	48%	56%	58%	57%	58%	52%	53%	0.29
Air pollution and environmental protection	35%	30%	32%	31%	34%	31%	25%	0.722
Diet and obesity	9%	20%	18%	20%	18%	18%	19%	0.17
Exercise	4%	10%	8%	7%	7%	10%	4%	0.117
Natural disasters	32%	20%	33%	25%	30%	27%	31%	0.01
Vaccination	26%	23%	18%	16%	15%	20%	26%	0.009
STDs	18%	14%	6%	8%	4%	3%	9%	<0.001
Depression	16%	25%	19%	20%	19%	19%	16%	0.274
Safe internet browsing	9%	5%	9%	5%	6%	8%	4%	0.188
Smoking	19%	22%	19%	24%	22%	26%	38%	0.012
Alcohol	6%	7%	6%	8%	10%	8%	7%	0.653
Contraception	6%	2%	5%	5%	1%	2%	1%	0.002
Safety in and around water	2%		1%	1%	0%	1%		0.256

STDs, Sexually transmitted diseases.

**Table 6 ijerph-18-08256-t006:** Chi-square tests regarding importance of public health risks by smoking status (importance defined as “Very important”, “Moderately important”, “Slightly important”, “Not important”, “Do not know/No answer”).

Public Health Risks	*p*-Value
Cancer	0.542
Diet and obesity	0.002
Exercise	0.025
Depression	0.320
Smoking	<0.001
Alcohol	0.267

Note: Table 6 presents *p*-values corresponding to Figure 5.

**Table 7 ijerph-18-08256-t007:** Most important public health risk (up to 3 answers per participant) by smoking status.

Public Health Risk	Non-Smokers	Smokers	DK/NA	*p*-Value
Cancer	55%	58%	47%	0.252
Diet and obesity	20%	14%	17%	0.011
Exercise	8%	7%	6%	0.760
Depression	18%	24%	25%	0.002
Smoking	28%	10%	17%	<0.001
Alcohol	7%	10%	8%	0.115

## Data Availability

The data presented in this study will be available upon request.

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
