# Peer review of "Perceived Importance of Public Health Risks in Greece: A Nationwide Survey of the Adult Population"

_ijerph, 2021, doi:10.3390/ijerph18168256_

Round 1

Reviewer 1 Report

All my previous concerns has been solved

Author Response

We thank the reviewer for advancing our manuscript through their critical approach.

Reviewer 2 Report

This is a much improved presentation and useful to understand the Greek situation.

Author Response

(The authors gave the same response as above.)

Reviewer 3 Report

This study adds to the literature on public health risks in Greece. It identifies the perceptions of the public to select health risks.

This cross sectional study has ensured that the sample is representative of the adult population of Greece. Verbal informed consent was obtained from each participant in the study. The statistical analysis of data has been completed. The results have been presented well..

The information collected on the populations perceptions versus the  actual public health risks in Greece have been presented in the discussion.

All information from this study can be used to inform public health policy in Greece

Author Response

(The authors gave the same response as above.)

Reviewer 4 Report

 This is a well-written paper. The author pointed out some suggestions. However, the difficulty of "perceived needs" lies in expressing people's needs rather than actual needs. Therefore, it is difficult to judge whether it is useful or policy development is prioritized or not, because it may have been done, but people still feel "needed"
In many high-risk factors, gender also shows great differences, but which one is more important than other factors, and it should be based on gender or education or various other factors will be another complicated issue. In this study, the important public health risks only STDs showed statistically significance, since Greece elderly felt unimportant but not the case of youth, which shows a large age gap between different age groups. However, smoking problems show similarities among different age groups, however, in reality, knowing is one thing to do is the other.
The survey sample size in this study is  too small, and the consent form for oral approval is a bit rough. For national-level reference papers, this may be useful for policy makers’ decision-making list, but for international readers, it needs to be more interesting outcomes or inspiring for others to consider as a model or further investigation.

Author Response

We thank the reviewer for their thought-provoking comments. Knowing the major health threats for the Greek population from national and European studies, through the present study we aimed to understand the perceptions of the public towards those threats, in order to prioritize and indicate the crucial areas for intervention that will ultimately improve the overall quality of public health in Greece.

This study searched how the Greek public perceives the importance of each public health risk included in the study. The statistical models used highlighted the role of age in the model of STDs, as the reviewer mentions, but also in the models of diet & obesity, alcohol, contraception, exercise, depression (in a smaller scale) and air pollution & environmental protection, in which an interaction of age and sex was observed. In the same context regarding the perception of the health risk caused by smoking, we showed that non-smokers were more likely to be more concerned for smoking as a public health risk compared to smokers, while older individuals were more likely to be more concerned compared to younger.

Considering that the maximum sampling error was 2% with a 95% confidence interval and that Greece’s population is around 11,000,000, the sample size used in the current study was adequate. Furthermore, the sample was representative of the adult Greek population according to age, sex and area of residence. This information is now added in the methods section of the manuscript.

The value of this paper rests on the fact that what the public perceives as an important health risk may differ from the major public health risks identified by scientific evidence. The gap between the two (perceived vs evidence-based) defines the areas policy makers need to focus on. The present study may be specific for the Greek public however the concept can be applied in any other population. To the authors’ knowledge, this is the first study in Greece as well as in other European countries, to present the public’s perception on a comprehensive list of public health risks and identify the top 3 that are considered the most important, which in order of importance are road accidents, cancer and air pollution & environmental protection. Additionally, “reverse reading” of the results helps identify gaps in awareness and education, highlighting a new field for the public health scientists to delve in and for campaigns to focus on, tailored to the current needs of the Greek Society.

Round 2

Reviewer 4 Report

Thanks for the authors' revised and explanation,  overall the paper's value has  proved for the further policy making and reference to further usage. In addition, the reviewer does agree the perception gap between the policy makers and general population should be focus on in order to make the better quality of health care.